# Phase Transition and Point Defects in the Ferroelectric Molecular Perovskite (MDABCO)(NH_4_)I_3_

**DOI:** 10.3390/ma16237323

**Published:** 2023-11-24

**Authors:** Francesco Cordero, Floriana Craciun, Patrizia Imperatori, Venanzio Raglione, Gloria Zanotti, Antoniu Moldovan, Maria Dinescu

**Affiliations:** 1Istituto di Struttura della Materia-CNR (ISM-CNR), Area della Ricerca di Roma—Tor Vergata, Via del Fosso del Cavaliere 100, I-00133 Rome, Italy; floriana.craciun@ism.cnr.it (F.C.); patrizia.imperatori@ism.cnr.it (P.I.); 2Istituto di Struttura della Materia-CNR (ISM-CNR), Area della Ricerca di Roma 1, Via Salaria, Km 29.300, Monterotondo Scalo, I-00015 Rome, Italy; venanzio.raglione@ism.cnr.it (V.R.); gloria.zanotti@ism.cnr.it (G.Z.); 3National Institute for Laser Plasma and Radiation Physics (NILPRP), 077125 Magurele, Romania; antoniu.moldovan@inflpr.ro (A.M.); maria.dinescu@inflpr.ro (M.D.)

**Keywords:** molecular ferroelectrics, organic perovskites, anelasticity, point defects complexes

## Abstract

We measured the anelastic, dielectric and structural properties of the metal-free molecular perovskite (ABX${}_{3}$) (MDABCO)(NH${}_{4}$)I${}_{3}$, which has already been demonstrated to become ferroelectric below $T_{C}=$ 448 K. Both the dielectric permittivity measured in air on discs pressed from powder and the complex Young’s modulus measured on resonating bars in a vacuum show that the material starts to deteriorate with a loss of mass just above $T_{C}$, introducing defects and markedly lowering $T_{C}$. The elastic modulus softens by 50% when heating through the initial $T_{C}$, contrary to usual ferroelectrics, which are stiffer in the paraelectric phase. This is indicative of improper ferroelectricity, in which the primary order parameter of the transition is not the electric polarization, but the orientational order of the MDABCO molecules. The degraded material presents thermally activated relaxation peaks in the elastic energy loss, whose intensities increase together with the decrease in $T_{C}$. The peaks are much broader than pure Debye due to the general loss of crystallinity. This is also apparent from X-ray diffraction, but their relaxation times have parameters typical of point defects. It is argued that the major defects should be of the Schottky type, mainly due to the loss of (MDABCO)${}^{2+}$ and I${}^{-}$, leaving charge neutrality, and possibly (NH${}_{4}$)${}^{+}$ vacancies. The focus is on an anelastic relaxation process peaked around 200 K at ∼1 kHz, whose relaxation time follows the Arrhenius law with τ${}_{0}$ ∼ $10^{-13}$ s and E≃0.4 eV. This peak is attributed to I vacancies (V${}_{X}$) hopping around MDABCO vacancies (V${}_{A}$), and its intensity presents a peculiar dependence on the temperature and content of defects. The phenomenology is thoroughly discussed in terms of lattice disorder introduced by defects and partition of V${}_{X}$ among sites that are far from and close to the cation vacancies. A method is proposed for calculating the relative concentrations of V${}_{X}$, that are untrapped, paired with V${}_{A}$ or forming V${}_{X}$–V${}_{A}$–V${}_{X}$ complexes.

## 1. Introduction

New organic and hybrid metal–organic ferroelectrics have been synthesized in the last few years, and are filling the gap between the polymer and the oxide ferroelectrics in terms of piezoelectric properties. In at least one case, they are even superior to the properties of PbZr${}_{1-x}$Ti${}_{x}$O${}_{3}$ (PZT) [1], and present advantages in terms of simplicity and cost effectiveness of preparation and flexibility of use. In fact, these materials can be prepared with purely chemical methods and deposited as films on surfaces of any shape. One major drawback is their limited thermal resistance, due to their weaker bonds and the volatility of the organic molecules at moderate temperatures.

The present record of piezoelectric response among metal–organics belongs to the hexagonal perovskite (TMFM)${}_{x}$(TMCM)${}_{1-x}$CdCl${}_{3}$ with $d_{33}=$ 1500 pC/N [1], but several molecular ferroelectrics have already been studied. Review articles on these materials have been published [2]; these studies focus on their piezoelectric properties and applications [3], but also on the broader perspective of hydrogen-bonded ferroelectrics [4] and metal–organic compounds [5,6], from the point of view of the multiferroic properties [7,8] and considering the possible applications [6,9].

A piezoelectric response is possible in non-centrosymmetric materials that are not ferroelectric, but it is generally much stronger in the ferroelectric state [10]. Many molecular ferroelectrics are hexagonal or layered in the paraelectric phase, but in principle, a cubic paraelectric phase is the most favourable for obtaining large piezoelectric responses below $T_{C}$, thanks to the reduced anisotropy, which allows for more orientations of the spontaneous polarization, as in the well-known perovskite oxides BaTiO${}_{3}$, PbZr${}_{1-x}$Ti${}_{x}$O${}_{3}$, etc. The first cubic perovskite of this type is the metal-free (MDABCO)(NH${}_{4}$)I${}_{3}$ [11], which is composed of corner-sharing I${}_{6}$ octahedra with NH${}_{4}$ in the centre and MDABCO molecules in the interstices among the octahedra. The MDABCO${}^{2+}$ cation is obtained from the roughly spherical DABCO = N${}_{2}$(C${}_{2}$H${}_{4}$)${}_{3}$ by attaching a methyl (CH${}_{3}$) group to one N in order to induce an electric dipole moment. Ferroelectricity, with a Curie temperature of 446 K, is due to the alignment of the dipole moments of these cations along a $\langle 111\rangle$ direction and their concomitant displacement along the same direction [12,13]. The piezoelectric constant of (MDABCO)(NH${}_{4}$)I${}_{3}$ is $d_{33}=$ 14 pC/N, which is enhanced to 63 pC/N upon substitution of the methyl with an amino group in MDABCO [14].

This molecular ferroelectric has also been studied via computational methods. Density functional theory calculations have revealed that the spontaneous polarization arising from the alignment of the dipole moment of MDABCO${}^{2+}$ along the $\langle 111\rangle$ axes is much amplified by a displacement of MDABCO${}^{2+}$ ions along the same axis [12,13]. Phase-field simulations have also been performed in order to derive a phase diagram and piezoelectric and dielectric properties as a function of strain, which are useful for enhancing such properties in thin films through epitaxial strain [15].

Further characterisation of (MDABCO)(NH${}_{4}$)I${}_{3}$ concerns its thermoelectric properties [16,17] and the static mechanical properties evaluated via nanoindentation and high-pressure [18,19]. It has also been demonstrated that this material can be prepared via mechanosynthesis [20], and the effects of various substitutions have been studied [14,21].

We present anelastic and dielectric spectroscopy measurements of (MDABCO)(NH${}_{4}$)I${}_{3}$ on samples of pressed powder, providing information on the process of thermal decomposition that already occurs close to $T_{C}$. The defects formed in this manner produce peaks in the elastic energy loss versus temperature, which are analyzed in terms of the mobility of iodine vacancies that can be trapped by cation vacancies.

## 2. Materials and Methods

### 2.1. Powders

(MDABCO)(NH${}_{4}$)I${}_{3}$ has been synthesized as reported in [18], and in detail:

First, 1.27 g (5 mmol) of MDABCOI (synthesized according to [22]) and 0.74 g of NH${}_{4}$I (5 mmol) were dissolved in 7.5 ml of H${}_{2}$O, 2.5 ml HI (57%) and 1 ml of H${}_{3}$PO${}_{3}$. The reaction mixture became opalescent, and H${}_{2}$O was added dropwise until it turned transparent again. It was kept in an ice-bath for one hour and then at room temperature for several hours, during which a microcrystalline solid precipitated. It was isolated by removing the supernatant with a pipette, and air-dried at 50 ${}^{\circ }$C on filter paper.

### 2.2. TGA

Thermogravimetric analysis was performed with a TGA/DSC2 apparatus (Mettler Toledo) at temperaturee ranging from 298 to 473 K at scan rate of 5 K/min and 2 K/min under nitrogen.

### 2.3. Bulk Samples

The bulk samples were obtained by pressing the powder into rectangular dies with dimensions 40×6 mm${}^{2}$ and circular dies of 13 mm diameter, for a few minutes, obtaining bars and discs with thicknesses of 0.6–0.9 mm. The applied pressures were 0.29 GPa for bars B1 and B2 and 0.37 GPa for B3, 0.75 GPa for disc D1. The average density of the bars was 2.05±0.04 g/cm${}^{3}$, but, due to the difficulty of uniformly distributing the powder on the bottom of the die, it was inhomogeneous along their length. This was evident from the nonuniform color along their length and may explain the large differences in the initial values of *E* in the three tested bars.

### 2.4. Anelastic Spectroscopy

The bars were suspended and fixed on thin thermocouple wires with drops of Ag paint. An electrode was placed close to the centre of the bar to electrostatically excite their flexural resonance modes at frequency *f*, as described in Ref. [23]. The capacitance between sample and electrode was inserted into a circuit resonating at ∼13 MHz, whose frequency was modulated by the sample vibration at 2f. The demodulated and rectified signal was detected with a lock-in amplifier locked at 2f. The Young’s modulus was obtained from the resonating frequency of the first mode as [24]
$$E=\rho {\left(\frac{f_{1}l^{2}}{1.028t}\right)}^{2}$$
where *l*, *t*, ρ are the sample’s length, thickness and density, which usually vary much less than *E* with temperature. Then, the temperature dependence of the Young’s modulus was deduced from $E/E_{0}=$${\left(f/f_{0}\right)}^{2}$, where the reference $f_{0}$ and $E_{0}$ are the resonance frequency and modulus, which were chosen as the starting values at the first measuring cycle. The elastic energy loss coefficient, $Q^{-1}=$$E^{\prime \prime }/E^{\prime }$, was measured from the width of the resonance peak or the decay of free oscillations.

### 2.5. Dielectric Spectroscopy

The dielectric permittivity $\varepsilon =\varepsilon^{\prime }+i\varepsilon^{\prime \prime }$ was measured with a HP4284A LCR meter with a four-wire probe during heating and cooling at 1.5–2 K/min in a modified Linkam HFS600E-PB4 stage. The cell was not perfectly air-tight, so during cooling the external humidity could penetrate and condense, leading to subsequent liquefaction and extrinsic dielectric anomalies during heating above 270 K, as in curve 2 of Figure 4.

### 2.6. Films

Films of (MDABCO)(NH${}_{4}$)I${}_{3}$ on 1×1 cm${}^{2}$ ITO/glass substrates were prepared by drop-casting a 10-fold diluted solution of precursors prepared as described in Section 2.1. Two drops were deposited on the substrate placed over a hot-plate at 80 ${}^{\circ }$C, and after the resulting layer dried, two more layers were added.

### 2.7. AFM, PFM

Surface morphology imaging was carried out with a commercial AFM (XE100, Park Systems, Suwon, Republic of Korea) in noncontact mode using ACTA tips (Applied NanoStructures, Mountain View, CA, USA). The PFM tests were performed on the same microscope using Pt-coated tips (NSC36Pt, Mikromasch—Innovative Solutions Bulgaria, Sofia, Bulgaria) to apply the local electric field and to record the material’s mechanical response. The out-of-plane piezoelectric response of the material was demodulated using an external lock-in amplifier (SR830, Stanford Research Systems, Sunnyvale, CA, USA).

## 3. Results

### 3.1. Anelastic Spectra

Figure 1 presents the normalized Young’s modulus $E/E_{0}$ and elastic energy loss $Q^{-1}=$$E^{\prime \prime }/E^{\prime }$ curves of bars B1–B3 during heating and cooling cycles in a high vacuum. All curves were obtained by exciting the fundamental flexural resonance with an initial frequency at room temperature of 0.8–1.0 kHz. The curves were normalized by dividing them by the initial modulus $E_{0}$, which was 11 GPa for B1, 6.6 GPa for B2 and 7.4 GPa for B3. The variability of the initial Young’s modulus may be attributed to nonhomogeneous density along the bars, which is evident from the nonuniform color observed immediately after their extraction from the press.

During the initial heating (curves 1, 4 and 9), all the $E\left(T\right)$ curves started softening almost linearly, as is usual in normal solids, in which the same anharmonicities responsible for thermal expansion also cause softening [25]. The slight relative stiffening above ∼380 K seemed to be an extrinsic effect, possibly caused by the loss of the solvent of the Ag paint of the electrode in the centre of the sample. In fact, the samples were measured immediately after applying the Ag electrode (see Methods). The samples then remained stable, as demonstrated by the perfectly reproducible thermal cycles on B2, where 400 K was not exceeded (curves 2 and 3). Heating through $T_{C}=446\pm 3$ K, which coincided with $T_{C}$ reported for single crystals [11], caused a steep drop of *E*, observed with perfect reproducibility on the three samples (curves 3, 4 and 9). Sample B1 broke immediately after passing this temperature, while the other samples did not, but were evidently deteriorated by the heat. In fact, heating of sample B3 was immediately stopped after the negative step in *E* (curve 4), and during the subsequent cooling, $T_{C}$ was depressed to 389 K. The anelastic spectrum thus became reproducible during heating and cooling, except for a thermal hysteresis of 20 K of the depressed $T_{C}$ (curves 5–8).

If heating was extended beyond the initial $T_{C}$ (curve 9), further softening occurred, which was not recovered during cooling (curve 10). Reaching 500 K resulted in ∼50% unrecoverable softening and a depression of $T_{C}$ down to 307 K. The decrease of $T_{C}$ and *E* was evidently caused by thermal decomposition beyond ∼450 K, presumably through Schottky defects, like loss of (MDABCO)I${}_{2}$, with formation of vacancies of MDABCO and iodine. These defects depress $T_{C}$ and weaken the lattice.

The elastic energy loss $Q^{-1}$ in the initial state was relatively low—below room temperature–and soon rose above 0.01 at higher *T* (B2). After heating above $T_{C}$, in addition to the permanent softening and lowering of $T_{C}$, at least three new peaks appeared in the $Q^{-1}$ curves, which were labelled P1–P3. Their intensities correlated with the maximum temperature reached, the magnitude of the permanent softening and the decrease of $T_{C}$, proving that they were caused by the defects introduced by the partial thermal decomposition. Indeed, Figure 2 shows that these peaks, including the rising background in the virgin state, shifted to higher *T* when measured at higher frequency, meaning that they are all thermally activated. The steps at $T_{C}=389$ K in sample B3 and $T_{C}=307$ K in sample B1 represent the low-temperature tail of the relaxation of domain walls, which disappears during the high-temperature phase. Peaks P1 (only partially visible) and P2 are stable and can be measured reproducibly during cooling and heating. Instead, the $Q^{-1}\left(T\right)$ curve above 300 K changed after four days in a vacuum; in place of the single P3 peak, there were two, one much smaller at slightly lower temperature and one of about the same intensity as P3, but broader, which shifted at a higher temperature of 50 K (not shown in Figure 2).

### 3.2. Dielectric Spectra

Figure 3 presents the real part and dielectric losses of disc D1 measured in air. The dielectric measurements on these samples of pressed powder were affected by the presence of intense Maxwell–Wagner relaxations from free charges, possibly of intergrain origin, and by poor adhesion of the Ag electrodes to the sample surface. The latter caused small jumps in $\varepsilon^{\prime }$, as shown in Figure 5 at 470 K. For this reason, rather than the Curie–Weiss peak in $\varepsilon^{\prime }$ observed in single crystals at $T_{C}≃446$ K [11], here, an anomaly appears at $T_{C}$ in the free charge relaxation (heating curves in Figure 3). Correspondingly, the dielectric losses are very high. As was also found in the anelastic experiments, heating at 1.3 K/min up to 490 K in air causes partial thermal decomposition and depresses $T_{C}$ down to 310 K. This effect is fully consistent with that observed in the anelastic measurement up to 500 K in a high vacuum, resulting in $T_{C}=$ 308 K, and suggests that there is not much difference in the loss of material above $T_{C}$ in air or vacuum (see also Figure 9 later on).

The left panel of Figure 4 presents $\varepsilon^{\prime }$ of D1 measured at 1 MHz during various thermal cycles. Only a decrease is observed during the initial cooling from a temperature lower than $T_{C}$. Curves 2 and 3 correspond to those at 1 MHz in Figure 3. In curve 4 (which is perhaps lower than curve 3 because of a partial detachment of the electrode), cooling is extended to lower temperature, so that it is more evident that the small step at the original $T_{C}$ becomes sharper and spiked and is followed by dielectric stiffening, parallel to the elastic stiffening; ($\varepsilon^{\prime }$ must be compared with the compliance 1/E).

The analogy between the dielectric and elastic susceptibilities is more evident in the right panel, which shows $\varepsilon^{\prime }$ of a piece of bar B3 measured on heating and cooling, after the anelastic measurements 5–8 of Figure 1. Here, the spike is absent and there is only a decrease of $\varepsilon^{\prime }$ from the high- to the low-temperature phase, at the same temperatures of the elastic steps, whose temperatures are indicated by vertical lines.

The losses below room temperature are initially featureless and decrease to <0.001 at 100 K, similarly to the anelastic losses. In order to check for counterparts of the anelastic peaks P1 and P2, Figure 5 presents the dielectric spectrum of disc D1 after 490 K has been reached. The transition has been depressed to 305 K, indicating that the sample is in a state similar to that of bar B1 in Figure 2. The thermally activated maximum around 200 K in tanδ seems compatible with P2, but the steep background makes any quantitative comparison or analysis difficult.

### 3.3. XRD and TGA

Figure 6 presents the X-ray diffractograms of the as-prepared powder of (MDABCO) (NH${}_{4}$)I${}_{3}$ and of bars B1–B3 after the anelastic measurements of Figure 1. There is a perfect correlation between the degradation of the spectra, in the sequence powder, B2, B3, B1, and the lattice disorder resulting from thermal decomposition at increasing maximum temperatures.

Figure 7 presents the normalized TGA curves measured by heating the powder in a Al${}_{2}$O${}_{3}$ crucible at 2 K/min and 5 K/min in N${}_{2}$. Two steps are observed: the first, with an onset around 390 K, should be the loss of water, while a second, with an onset around 435 K, may be due to the loss of cations and anions resulting from the formation of Schottky defects.

### 3.4. AFM and PFM

In order to verify the existence of the piezoelectric response, a film of (MDABCO)(NH${}_{4}$)I${}_{3}$ was deposited from solution on an ITO/glass substrate for AFM and PFM analysis (Figure 8). The deposition parameters were not sufficiently optimised because the optical images from AFM (∼480 μm ×360 μm) show that the surfaces are inhomogeneous. Some zones have valleys; others have a rough appearance. Both types of zones show variation in the thickness of micrometers, as estimated by refocusing the optical image. The topographical AFM images show the presence of some agglomerations of grains, which are partially embedded in a glassy matrix. The grains have dimensions of up to 1 μm.

The PFM images were obtained by applying an AC voltage of about 5 V due to the large thickness of the films. Some grains have a piezoelectric response, which is demonstrated by the contrast in the images of PFM amplitude and PFM phase in correspondence with some grains.

## 4. Discussion

### 4.1. Thermal Decomposition

From these measurements, one deduces that polycrystalline (MDABCO)(NH${}_{4}$)I${}_{3}$ suffers thermal decomposition in a vacuum just above 445 K, as demonstrated by the lowering of $T_{C}$, the appearance of intense thermally activated processes in the anelastic and dielectric spectra and the degradation of the XRD spectra.

Figure 9 shows the lowering of $T_{C}$ as a function of $T_{\max}$, the maximum temperature reached during the anelastic measurement in high vacuum or dielectric measurements in air. The points at the top correspond to the initial state and are obtained by setting $T_{\max}=T_{C}$. Both sets of data follow the same line, so that there is no visible effect of the atmosphere on the process of thermal degradation. Single crystals have a reduced exposed surface with respect to polycrystals and should follow a higher curve. In fact, the crosses are the reported $T_{C}$ during heating and cooling of single crystals measured with DSC and SHG [11], where $T_{\max}$ is set to the upper values of the abscissas in the respective figures (hence, it might be higher). Considering that in our polycrystals, the thermal hysteresis of reduced $T_{C}$ does not exceed 20 K (curves 5–8 of Figure 1), it seems likely that the hysteresis of >50 K reported in the single crystals is rather due to the fact that during cooling, the samples were degraded.

Further evidence of mass loss above 440 K in <$10^{-5}$ mbar includes a marked rise in pressure above that temperature during continuous pumping in the anelastic measurements, and a yellow/brown colouration of the quartz tube enclosing the sample holder. This is typical of the deposition of iodate compounds.

The major mechanism of degradation of perovskites ABX${}_{3}$ at high temperature is the formation of Schottky defects, namely pairs or complexes of anion X and cation A or B vacancies that leave the neutral total charge unchanged according to the formal valence of the ions. These vacancies can then migrate into the bulk, with the combined effects of depressing the temperatures of the structural transitions, softening the lattice and producing anelastic and dielectric relaxation, if the vacancies or their complexes have electric or elastic dipoles. Both effects are very strong in the present measurements.

It should be remarked that (MDABCO)(NH${}_{4}$)I${}_{3}$ begins to deteriorate at $T_{C}$ because it has a particularly high $T_{C}$, close to the onset of decomposition of any organic material. This does not make it less suitable than other organic and metal–organic ferroelectrics for applications.

### 4.2. Elastic and Dielectric Anomalies at $T_{C}$

The ferroelectric transition of (MDABCO)(NH${}_{4}$)I${}_{3}$ is reported to be of the species 432R3 [26] from the paraelectric cubic P432 phase to the ferroelectric rhombohedral R3 phase [11]. The dielectric permittivity of single crystal (MDABCO)(NH${}_{4}$)I${}_{3}$ has a Curie–Weiss peak, whose amplitude is strongly dependent on frequency: it passes from 15,000 at 20 Hz to 100 at 1 MHz [11], as a result of the slow reorientational dynamics of the polar (MDABCO)${}^{2+}$ molecules. The major contribution to spontaneous polarization, however, is not due to the freezing of the molecular polar axis along one of the cubic $\langle 111\rangle$ directions, but to a displacement of (MDABCO)${}^{2+}$ along the same direction [11].

We were not able to resolve the Curie–Weiss peak observed in single crystals due to the high conductivty, presumably of intergranular origin, but there are several indications that our material was of good quality: all the powder XRD peaks corresponded with those reported in the literature, and $T_{C}$ measured on the pressed powder is identical to that in single-crystal. In addition, PFM-detected piezoelectric activity at the subgrain level. In particular, the close correspondence of $T_{C}$ to the value found in single crystals indicates that the samples are of good quality, in view of its large shifts after defects have been introduced.

Unlike the dielectric susceptibility, the elastic response is unaffected by free charges, and the elastic anomaly in normal ferroelectrics is a steplike softening below $T_{C}$ [27] of piezoelectric origin [28]. This is in opposition to what is observed in the present case, in which all three samples soften above $T_{C}$. This behaviour can be explained if the transition is not properly ferroelectric with the polarisation as a primary order parameter, but is mainly due to the loss of the free rotation of the molecular cation MDABCO${}^{2+}$. From this point of view, the step in the modulus is nothing other than the softening from the coupling between strain and the molecular rotation mode, which is frozen below $T_{C}$. Order–disorder transitions of the molecular orientations of this type are found in the metal–organic perovskites NH${}_{4}$-Zn(HCOO)${}_{3}$ [29] and TMCM-MnCl${}_{3}$ [30,31], also improper ferroelectric, and MAPbI${}_{3}$ [32] and FAPbI${}_{3}$ [33] (tetragonal-to-orthorhombic transitions). As discussed in the latter cases, part of the stiffening in the low-temperature phase may be due to the formation of stronger H bonds of the ordered molecules with the surrounding halide octahedra.

When$\;T_{C}$ is lowered by defects, the dielectric anomaly becomes more evident (in opposition to the anelastic one; see Figure 3 and Figure 5), but it is not a Curie–Weiss peak, whose slope at low temperature should be higher than at high temperature. This fact, together with the softening rather than stiffening that occurs during the transition to the paraelectric phase, indicates that the transition is not of the Curie–Weiss type with the polarization as the order parameter; it is driven by something else, and the appearance of the electric polarization is a side effect.

### 4.3. Grain Boundary Relaxation

The mechanical loss $Q^{-1}$ of as-pressed (MDABCO)(NH${}_{4}$)I${}_{3}$ is relatively low but starts increasing considerably above room temperature (B1 in Figure 1), is thermally activated (Figure 2) and is repeatable up to 400 K. This increase in dissipation above room temperature should be due to grain boundary (GB) sliding. Grain boundary relaxation is usually considered relevant at temperatures higher than half the melting temperature [34], and it causes a very broad peak in the mechanical losses [35]. Our material begins to decompose slightly above 400 K, and therefore peak P4 may be caused by GB sliding. Indeed, organic polycrystals are used as model systems to study the anelasticity from GB sliding in rocks at room temperature rather than at ≫1000 K, since the respective melting temperatures pass from thousands to hundreds of kelvin [36].

The GB peak results from a very broad distribution of relaxation times, due to the distribution of sizes of the grains and of the degrees of coherency of their boundaries. In Al${}_{2}$O${}_{3}$ and MgO, it has been found that the low-temperature components of the GB peak disappear with annealing, which is interpreted as being caused by an increase in the degree of GB coherency with grain growth [35]. The same process may occur in our organic polycrystal: exceeding $T_{C}$ would not only cause a loss of material, but also anneal the GB structure formed during pressing at room temperature, explaining why in Figure 1 $Q^{-1}$ above room temperature in sample B3 is lower than in unannealed B2.

### 4.4. Point Defects Relaxations

At least three thermally activated peaks, P1–P3, appear when $T_{C}$ is depressed by partial decomposition, and they must be caused by newly created defects.

In defective perovskites ABX${}_{3}$ (Figure 10) the most abundant and mobile defects causing anelastic relaxation are the X vacancies (V${}_{X}$). When a V${}_{X}$ jumps to a nearest neighbour X position along an edge of a BX${}_{6}$ octahedron, the direction of its two nearest neighbour B atoms, at the centre of the octahedra, rotates by $90^{\circ }$, and therefore the local anisotropic strain (elastic dipole) also rotates. The elastic energies of these three types of elastic dipoles change upon application of uniaxial stress and therefore induce changes of their average populations. This results in a relaxation of the macroscopic strain due to the elastic dipoles with kinetics determined by their mean hopping time; this is called anelastic relaxation [24]. If periodic stress with angular frequency ω is applied, the continuous readjustment of these populations causes an additional retarded anelastic strain, and therefore an increment of the elastic compliance S=1/E given by $\Delta S=\Delta /\left(1+\iota \omega \tau \right)$, where the relaxation time generally follows the Arrhenius law
(1)$$\tau =\tau_{0}\exp\left(E/T\right)$$
with $10^{-15}\;$s <$\tau_{0}<$$10^{-12}$ s for point defects, and the relaxation strength is
(2)$$\Delta \propto \frac{n{\left(\Delta \lambda \right)}^{2}}{k_{B}T}\;,$$
where *n* is the defect population and Δλ represents the change in the elastic dipole after the jump/reorientation of the defect. Each process contributes to the losses as a Debye relaxation [24,37]
(3)$$Q^{-1}=\frac{S^{\prime \prime }}{S^{\prime }}=\Delta \frac{\omega \tau }{1+{\left(\omega \tau \right)}^{2}},$$
peaked at *T*, such that $\omega \tau \left(T\right)≃1$. Notice that isolated V${}_{X}$ do not have an electric dipole and therefore do not cause dielectric relaxation, while pairs of cation and anion vacancies have both elastic and electric dipoles.

Even though peaks P1–P3 are considerably broader than Debye relaxations, they are clearly caused by well-defined defects with quite different activation barriers *E*. It would be tempting to make parallels with the anelastic relaxation spectra of other defective perovskites, like O deficient SrTiO${}_{3-\delta }$ [38] and partially decomposed (TMCM)MnCl${}_{3}$ [31], but the present situation is different. Oxide perovskites are quite stable compounds and may loose only O atoms at high temperature in a reducing atmosphere. The charge compensation in SrTiO${}_{3-\delta }$ from the loss of O${}^{2-}$ anions can be achieved by the reduction of 2δ Ti${}^{4+}$ cations to Ti${}^{3+}$. The resulting defects are V${}_{O}$ and small polarons and the anelastic spectra of SrTiO${}_{3-\delta }$ and BaTiO${}_{3-\delta }$ show peaks due to their hopping, with clearly distinct peaks for vacancies that are isolated and paired [38,39].

In halide perovskites heated close to the decomposition temperature, it is unlikely that only anion vacancies are formed, since the organic cation is volatile and the prevalent mechanism of decomposition is expected to be the formation of Schottky defects, namely neutral pairs or complexes of cation and anion vacancies. In (TMCM)MnCl${}_{3}$, it was assumed that an equal concentration of TMCM${}^{+}$ and Cl${}^{-}$ was formed [31] for two reasons: (i) A = TMCM is organic and more volatile than inorganic B = Mn; (ii) vacancies on the B-site of perovskites are rare, though they may be created under particular circumstances [40,41,42]. A reason for the tendency to lose A rather than B ions is that the BX${}_{6}$ octahedra are clearly the stable backbone of the lattice, because the B-X bonds are shorter than the A–X ones and are therefore stronger, especially when they involve a greater charge (B${}^{4+}$ and A${}^{2+}$ or B${}^{2+}$ and A${}^{+}$). That the B–X bonds are much stronger than the A–X ones is demonstrated by the common tilting transitions of the octahedra upon cooling [43]. In fact, during cooling, the weaker and more anharmonic A–X sublattice contracts more than the rigid network of BX${}_{6}$ octahedra, and the octahedra, unable to compress, rigidly tilt [44]. In the present case, both the conditions of smaller volatility of B and stronger B–X bonds are impaired: not only are both cations organic, but B${}^{2+}$ has a greater charge than A${}^{+}$. It is therefore possible that in (MDABCO)(NH${}_{4}$)I${}_{3}$ the loss through Schottky defects of NH${}_{4}^{+}$ with one I${}^{-}$ is not negligible with respect to the loss of MDABCO${}^{2+}$ with two I${}^{-}$. Yet, only jumps of V${}_{X}$ are expected to occur over barriers E<0.5 eV, small enough to cause anelastic relaxation below room temperature. In fact, the jumps of V${}_{A}$ and V${}_{B}$ are $\sqrt{2}$ times longer than the octahedron edge and must occur with the participation of another V${}_{X}$ or cation vacancy of the other type. As a consequence, vacancies V${}_{X}$ are far more mobile than V${}_{A/B}$, and this is true not only in perovskite oxides but also in the metal-organic halide perovskites for photovoltaic applications [45].

Peaks P1–P3 should therefore be attributed to different types of jumps of the I${}^{-}$ vacancies V${}_{X}^{+}$ among quasistatic MDABCO${}^{2+}$ vacancies V${}_{A}^{2-}$ and possibly NH${}_{4}^{+}$ vacancies V${}_{B}^{-}$. The major mobile defects that should be considered are V${}_{X}^{+}$ that are: (i) isolated, (ii) paired with V${}_{A}^{2-}$ and (iii) paired with V${}_{B}^{-}$; (iv) form neutral V${}_{A}^{2-}-2$V${}_{X}^{+}$ complexes. Pairs of V${}_{X}^{+}$, such as those in perovskite oxides, are unlikely to form in the presence of cation vacancies, which provide stronger binding energies of an electrostatic nature.

We are not aware of studies on the mobility of V${}_{X}$ in the presence of cation vacancies in perovskites, but we assume that the jumps within the sites that are the nearest neighbour to the cation vacancies are faster than those in the unperturbed lattice, because the missing cation certainly lowers the hopping barrier. It is less obvious to establish which cation vacancy provides the easiest environment for V${}_{X}$ hopping, whether the octahedron centred on V${}_{B}$ or the cuboctahedron centred on V${}_{A}$. We assume that V${}_{B}$ lowers more than V${}_{A}$ the hopping barrier for V${}_{X}$, because of the shorter B–X bond. The influence of V${}_{B}$ should therefore be more sensible than that of V${}_{A}$.

Based on the above considerations, as a first guess, we assume that P1 is due to jumps of V${}_{X}$ around V${}_{B}$, or equivalently to the reorientation of V${}_{B}$–V${}_{X}$ pairs, and P2 to V${}_{A}$–V${}_{X}$ pairs. The possible V${}_{X}$–V${}_{A}$–V${}_{X}$ complex, presumably with V${}_{X}$ on opposite sides of V${}_{A}$, requires a two-step reorientation, and the stronger lattice relaxation around an in-line V${}_{X}$–V${}_{A}$–V${}_{X}$ complex suggests a higher barrier, due the unfavourable intermediate state.

Finally, one should consider the plethora of jumps between non-equivalent sites, e.g., between first and second neighbours to a cation vacancy. These jumps would contribute both to the broadening of the major relaxation peaks and to the background. In fact, if one considers the relaxation between states differing in energy of *A*, then the relaxation strength is reduced by the factor [46].
(4)$$\Delta \to \Delta /\cosh^{2}\left(A/2k_{B}T\right)\;,$$
which below $k_{B}T∼A$ transforms the 1/T divergence of Δ into an exponential decrease to 0 (Figure 11).

Therefore, jumps between strongly inequivalent positions, like those required to form and separate a defect pair, produce relaxation processes distinct from those cited above, and with a much depressed intensity. This type of relaxation between inequivalent states is hardly distinguishable in SrTiO${}_{3-\delta }$ with δ≲0.01, where a small but distinct peak can be attributed to the partial dissociation of pairs of V${}_{O}$ [38].

We should also take into account the temperature dependence of the relaxation strengths Equation (Equation 2), which are proportional to $n\left(T\right)/T$, where *n* is the relative concentration of the defect. This fact should help distinguish the population of isolated vacancies from those of the defect complexes, since the first rises with temperature while the others decrease due to the thermal dissociation.

The considerable width of the peaks with respect to single Debye relaxations is justified by the presumably high concentration of defects and by the softness of the lattice, which result in large local deviations of the bond lengths and angles from the ideal lattice.

### 4.5. Two Possible Scenarios for P2

Our attention will focus on peak P2, because it is clearly observable in both samples annealed at high temperature and is stable, while peak P3 is observable only at the highest content of defects and is not stable; of P1, only the high-temperature tail could be seen. A feature of P2 is the change of the temperature dependence of its intensity when passing from low- to high-defect concentration. The peak is maximum at $T_{1}$ and $T_{3}$ when measured at the low and high frequencies $f_{1}$ and $f_{3}$. If we define the ratio of the intensities at the temperatures
(5)$$r=\Delta \left(T_{3}\right)/\Delta \left(T_{1}\right)$$
then at low defect content (sample B3), it is r≃1, while at high content of defects (sample B1), it is r≃1.2. The rise of *r* with a rising defect concentration is large and can be explained by two concomitant mechanisms: (i) the increase in the lattice disorder, and therefore of the average asymmetry *A* in Equation (Equation 4); (ii) the increase with the temperature of the population of the defects producing P2. We will discuss the two possibilities separately, although they can be concomitant, in relation to our hypothesis that P2 is caused by the reorientation of V${}_{A}$–V${}_{X}$ pairs, and possibly also V${}_{X}$–V${}_{A}$–V${}_{X}$ complexes.

We first assume that the population of such defects depends little on temperature in the range in which P2 is observed, and therefore only a change of *A* in Equation (Equation 4) occurs. Figure 11 shows how the 1/T dependence of the relaxation strength Δ is depressed by increasing the average asymmetry *A* between relaxing states. The two values A=320 and 670 K are those of the fits of P2 at low and high defect concentrations, which yield r=1.007 and 1.23, respectively. The fits are obtained with the expressions
(6)$$\begin{array}{ccc} Q^{-1} & = & \frac{\Delta_{0}}{T\cosh^{2}\left(A/2T\right)}\frac{1}{{\left(\omega \tau \right)}^{\alpha }+{\left(\omega \tau \right)}^{-\beta }}\\ \tau & = & \tau_{0}\exp\left(E/T\right)\cosh\left(A/2T\right) \end{array}$$
which are a generalization of Equation (Equation 3) to include asymmetric broadening with α,β≤1 and energy asymmetry *A*. The factor in the expression of the relaxation time is due to the fact that $\tau^{-1}=\nu_{12}+\nu_{21}$ between states 1, 2 with energies ±A/2 and separated by a barrier $E_{1,2}=$E∓A/2. In this formula, the energies are expressed in kelvin, while they are actually $E/k_{B}$ and $A/k_{B}$, because in this manner their influence on the fitting curves is more transparent.

The fit of sample B3 includes the two adjacent peaks: P1 and that attributed to domain wall relaxation, plus a linear background. The parameters of interest in P2 are in the first column of Table 1:

The relaxation time extrapolated to infinite temperature, $\tau_{0}$, is typical of point defects, and the activation energy $E/k_{B}=$ 4730 K corresponds to 0.41 eV. This would be the barrier for the local motion of an iodine vacancy around a MDABCO vacancy. The effective activation energy for the long-range diffusion of I would be larger, because it would include the higher barrier for escaping from V${}_{A}$ and, as discussed above, the activation energy for hopping in the unperturbed lattice might be higher than around a cation vacancy. The peak is much broader than a single time relaxation, since α=β are definitely <1, and this justifies an average asymmetry *A* that is 6–7% of the activation energy.

The parameters of peak P1 cannot be reliably determined, but we mention that its relaxation time has $\tau_{0}≃$
$10^{-14}$ s and E≃0.15 eV. This values are, again, typical of point defects, but it is unlikely that an iodine vacancy can diffuse so fast in the unperturbed lattice, and we suggest that this is the barrier for hopping around a NH${}_{4}$ vacancy. In addition, at such a low temperature, all the V${}_{X}$ are trapped, as will be shown later.

The right panel of Figure 12 shows a similar fit of P2 with high defect contents (sample B1). The parameters corresponding to the continuous lines are in the second column of Table 1: the average site energy disorder, *A*, is doubled, which is reasonable, and the peak shape is much more asymmetric, with very different broadening parameters: α and β. With such broad shape, there is not much sensitivity to changes in the pair of parameters $\tau_{0}$ and *E*, and it is possible to fit with $10^{-14}$ s<$\tau_{0}<$$10^{-12}$ s accordingly, varying *E* within 3800 K – 4800 K. A similar remark holds for the previous fit.

While this pair of fits is compatible with the hypothesis that P2 is caused by the hopping of V${}_{X}$ around V${}_{A}$, the presence of various types of defects, confirmed by the presence of peak P1, calls for an analysis of the defect populations as a function of temperature.

### 4.6. Calculation of the Populations of Defects

In order to calculate the populations of the various types of defects, based on their respective energies and geometries, we will adapt the calculation of the mean occupation numbers $n_{i}$ from the grand partition function [47]
(7)$$Z=\prod_{i=1}^{N}\left(1+e^{\beta \left(\mu -E_{i}\right)}\right)$$
where, for each site *i* with energy $E_{i}$, 1 is the statistical weight for being unoccupied and the second term for being occupied; β=$1/k_{B}T$ and μ is the chemical potential, or Fermi energy in this case. The mean occupation numbers with a total of *M* occupied states are calculated as
(8)$$M=\frac{1}{\beta }\frac{\partial \ln Z}{\partial \mu }=\sum_{i=1}^{N}\frac{e^{\beta \left(\mu -E_{i}\right)}}{1+e^{\beta \left(\mu -E_{i}\right)}}=\sum_{i=1}^{N}n_{i}\;.$$

By grouping different types of sites and assigning different site energies, it is possible to obtain the occupations $n_{i}$ of coexisting defects configurations and calculate them by numerically solving the above implicit equation for μ [46,48].

In the present case, $n_{i}=1$ represents a V${}_{X}$ vacancy in site X${}_{i}$ and $n_{i}=0$ a regular X${}_{i}$ site. For simplicity, let us consider the case that only V${}_{A}^{2-}+2$V${}_{X}^{+}$ Schottky defects are formed. Then, there are $c_{X}=\delta$ iodine vacancies V${}_{X}^{+}$ and $c_{A}=\delta /2$ MDABCO vacancies V${}_{A}^{2-}$ per mole. The V${}_{A}$ are assumed to be static and isolated, so each of them has m=12 X nearest neighbour (nn) sites. In *Z*, we can group together the $c_{0}=3-mc_{A}$ normal sites with energy $E_{0}=$ 0 and those nn to a same V${}_{A}$ with binding energy $E_{k}=$ B< 0 so that, for *N* unit cells,
$$Z=\prod_{i=1}^{Nc_{0}}\left(1+e^{\beta \mu }\right)\prod_{k=1}^{Nc_{A}}{\left(1+e^{\beta \left(\mu -B\right)}\right)}^{m}=z_{0}^{Nc_{0}}z_{A}^{Nc_{A}}\;.$$

At this point, it is possible to introduce arbitrary conditions on the type of V${}_{A}$–V${}_{X}$ complexes by expanding and manipulating the sub-partition function $z_{A}$ of each set of *m* sites nn to a V${}_{A}$. From the polynomial expansion of $z_{A}$, we retain only the terms with up to two V${}_{X}$
$$z_{A}=1+me^{\beta \left(\mu -B_{1}\right)}+\frac{m}{2}e^{2\beta \left(\mu -B_{2}\right)}$$
setting $B_{1}<0$ as the energy of a V${}_{A}$–V${}_{X}$ complex and $B_{2}<0$ as that of a V${}_{X}$–V${}_{A}$–V${}_{X}$ complex. In addition, we suppose that the latter can only have V${}_{X}$ on opposite sides of V${}_{A}$, so that there are only m/2 such configurations. In this manner, Equation (Equation 8) becomes
$$\begin{array}{ccc} c_{X} & = & \frac{1}{\beta N}\frac{\partial \ln Z}{\partial \mu }=\frac{1}{\beta }\frac{\partial }{\partial \mu }\left[c_{0}\ln\left(1+e^{\beta \mu }\right)+c_{A}\ln\left(1+me^{\beta \left(\mu -B_{1}\right)}+\frac{m}{2}e^{2\beta \left(\mu -B_{2}\right)}\right)\right]=\\ & = & \left[c_{0}\frac{e^{\beta \mu }}{\left(1+e^{\beta \mu }\right)}+c_{A}\frac{me^{\beta \left(\mu -B_{1}\right)}+me^{2\beta \left(\mu -B_{2}\right)}}{1+me^{\beta \left(\mu -B_{1}\right)}+\frac{m}{2}e^{2\beta \left(\mu -B_{2}\right)}}\right] \end{array}$$
with
$$\begin{array}{ccc} c_{X} & = & \delta \\ c_{A} & = & \delta /2\\ c_{0} & = & 3-mc_{A}=3-m\delta /2 \end{array}$$
where, setting
$$\begin{array}{ccc} y & = & e^{\beta \mu }\\ w_{1} & = & e^{-\beta B_{1}}>1\\ w_{2} & = & e^{-\beta B_{2}}>1\;, \end{array}$$
we recognize the occupation numbers of isolated V${}_{X}$ and V${}_{A}$–V${}_{X}$ and V${}_{X}$–V${}_{A}$–V${}_{X}$ complexes as
(9)$$\begin{array}{ccc} n_{0} & = & \left(3-\frac{m}{2}\delta \right)\frac{y}{\left(1+y\right)}\\ n_{1} & = & \frac{\delta }{2}\frac{myw_{1}}{1+myw_{1}+\frac{m}{2}y^{2}w_{2}^{2}}\\ n_{2} & = & \frac{\delta }{2}\frac{my^{2}w_{2}^{2}}{1+myw_{1}+\frac{m}{2}y^{2}w_{2}^{2}} \end{array}$$

The chemical potential in *y* is determined by the condition
(10)$$\delta =n_{0}+n_{1}+n_{2}\;.$$

This is a third-degree polynomial equation in y, which is best solved numerically. Notice that, without the V${}_{X}$–V${}_{A}$–V${}_{X}$ complexes ($w_{2}=0$), the equation is of the second degree and $n_{0}$ and $n_{1}$ have a simple closed form. The inclusion of additional defects—for example, V${}_{B}$–V${}_{X}$ pairs—can be easily implemented in the same manner, but, with only two measurements of P2, it is not worth introducing new parameters. It is also not worth including the numerical solution of Equation (Equation 10) in the non-linear fitting procedure of the anelastic spectra, and therefore we simply identify a combination of binding energies $B_{1}$ and $B_{2}$ and reasonable values of δ, which roughly reproduce the observed temperature dependences of P2.

Figure 13 shows the occupation numbers calculated with $B_{1}=$−1500 K, $B_{2}=$−1300 K at three defect concentrations δ. In the low-*T* limit, all V${}_{X}$ are trapped and therefore $n_{2}=1$, but above 100 K, they start to detrap, each time forming a V${}_{A}$–V${}_{X}$ pair and an isolated V${}_{X}$, so that $n_{1}≃n_{0}.$ Finally, at $T>B_{1},B_{2}$, there is an equal probability of occupying all sites, and the populations depend only on the total number of trap and free sites. In the intermediate region, $n_{1}$ has a maximum, which shifts to higher *T* with increasing δ, and can therefore account for the behaviour of P2. In fact, passing from δ=0.01 to 0.1, the ratio *r* of the intensities of P2, Equation (Equation 5), passes from 1.04 to 1.21, which is compatible with Figure 12.

Therefore, we fitted P2 again at a high defect concentration using Equation (Equation 3) with A=0 and $\Delta_{0}\propto n_{1}\left(T,\delta =0.1\right)$. In practice, we fitted the intermediate $n_{1}\left(T\right)$ curve in Figure 13 with a rational expression (the blue line in the right panel of Figure 13), which we introduced in the final fitting expression for $Q^{-1}\left(T\right)$.

The dashed lines in Figure 12 are obtained in this manner with the following parameters: $\tau_{0}=$
$3.4\times 10^{-12}$ s, E= 3610 K (0.31 eV), α=0.15, β=0.6, and the resulting fit practically coincides with the previous one with constant $\Delta_{0}$ and A>0. This suggests that it is possible to fit peak P2 equally well, assuming intermediate situations with slightly different values of the binding energies to the cation vacancies and A>0.

We have therefore shown that peak P2 can be explained in terms of reorientation of V${}_{A}$–V${}_{X}$ pairs, corresponding to the local hopping of an iodine vacancy around a MDABCO vacancy, with an activation energy E=0.35±0.05 eV. The change in the temperature dependence of the P2 intensity, passing from low to high defect concentrations, can be explained both in terms of increasing lattice disorder (site energy asymmetry *A*) and with the presence of at least one other type of V${}_{X}$ trapping (V${}_{X}$–V${}_{A}$–V${}_{X}$ complexes) that compete with the V${}_{A}$–V${}_{X}$ pairs. The introduction of V${}_{B}$–V${}_{X}$ pairs, possibly giving rise to peak P1, could only improve the description of the anelastic spectra, but the present data, at only two defects concentrations and limited in the low temperature range, do not justify fittings with so many parameters.

## 5. Conclusions

We measured the complex Young’s modulus and dielectric permittivity of the ferroelectric organic perovskite (MDABCO)(NH${}_{4}$)I${}_{3}$ in polycrystalline form. The samples of pressed powder were measured during temperature cycles with increasing maximum temperature below and above the transition temperature $T_{C\;}$ in air and in a vacuum. Thermal decomposition began at $T_{C}$, as indicated by the decreased $T_{C}$ during cooling and the subsequent cycles, and by the concomitant appearance of intense anelastic and dielectric relaxation processes.

The observations are explained in terms of the formation of Schottky defects at high temperature, namely the loss of neutral complexes of anions X and cations A and B in perovskite ABX${}_{3}$, with the respective vacancies migrating into the bulk. The anelastic spectra are interpreted in terms of hopping of iodine vacancies V${}_{X}$ partially trapped by cation vacancies, V${}_{A}$ and V${}_{B}$, which are much less mobile. In particular, the two peaks of elastic energy loss starting from the lowest temperature are interpreted in terms of reorientation of V${}_{B}$–V${}_{X}$ and V${}_{A}$–V${}_{X}$ pairs. Of the first peak, only the high-temperature tail is observed, but the second is fitted taking into account the lattice disorder and the temperature dependence of the various populations of defects: isolated V${}_{X}$, V${}_{A}$–V${}_{X}$ pairs and V${}_{X}$–V${}_{A}$–V${}_{X}$ complexes. These populations are calculated with a simple method based on the Fermi–Dirac statistics.

## Figures and Tables

**Figure 1 materials-16-07323-f001:**
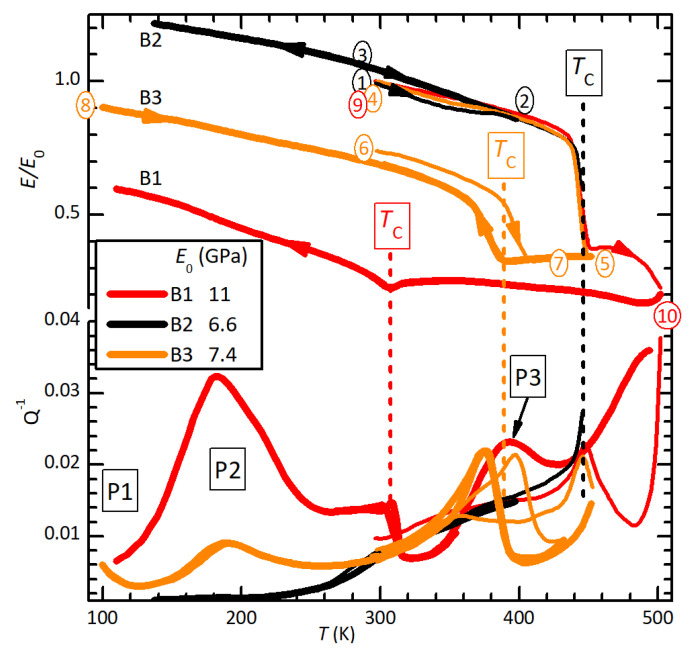
Normalized Young’s moduli $E/E_{0}$ and $Q^{-1}$ curves measured on samples B1–B3 during cooling (thick) and heating (thin) in a high vacuum, exciting the fundamental flexural resonance, initially at 0.8–1.0 kHz. Only splines through the experimental data are shown.

**Figure 2 materials-16-07323-f002:**
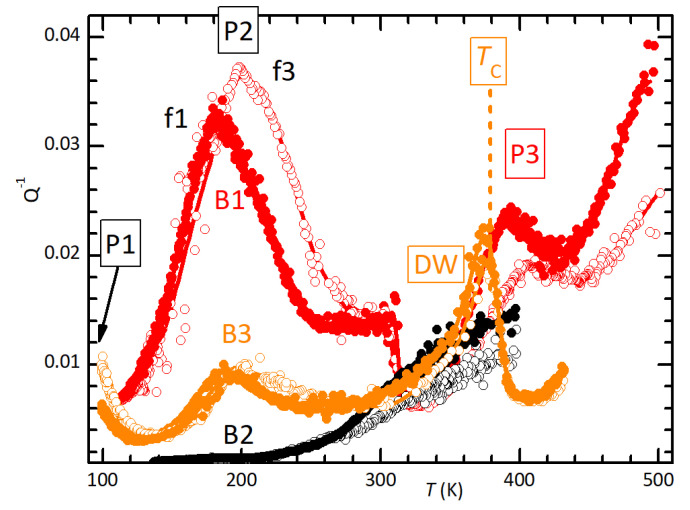
Anelastic spectra of samples B1–B3 measured exciting the 1st (filled symbols) and 3rd (open symbols) flexural resonances at 1.1 and 5.9 kHz (B1), 0.81 and 4.9 kHz (B2) and 0.93 and 5.4 kHz (B3). All measurements were taken during cooling and B3 both cooling and heating.

**Figure 3 materials-16-07323-f003:**
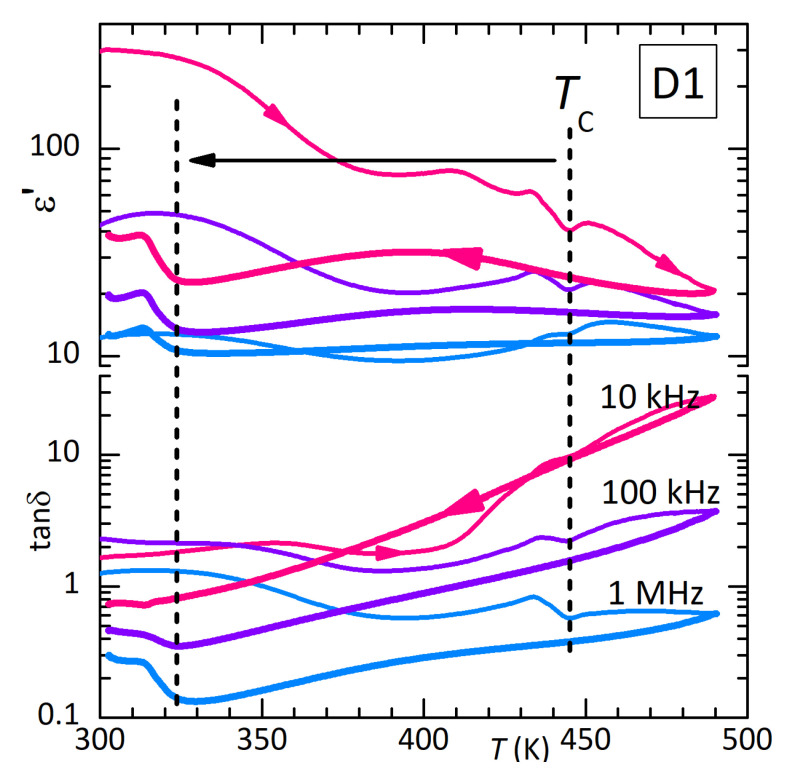
Dielectric permittivity measured in air on disc D1 at three frequencies during heating (thin lines) and subsequent cooling (thick lines). The partial thermal decomposition at the highest temperatures caused a decrease of $T_{C}$.

**Figure 4 materials-16-07323-f004:**
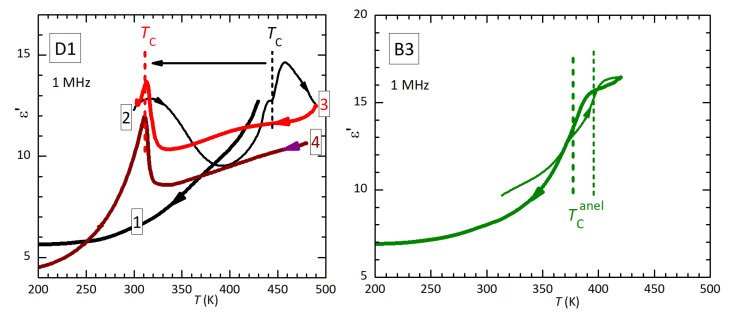
Dielectric permittivity measured at 1 MHz during heating (thin lines) and cooling (thick lines) on disc D1 and a piece of bar B3. Curves 2 and 3 correspond to those at 1 MHz in Figure 3.

**Figure 5 materials-16-07323-f005:**
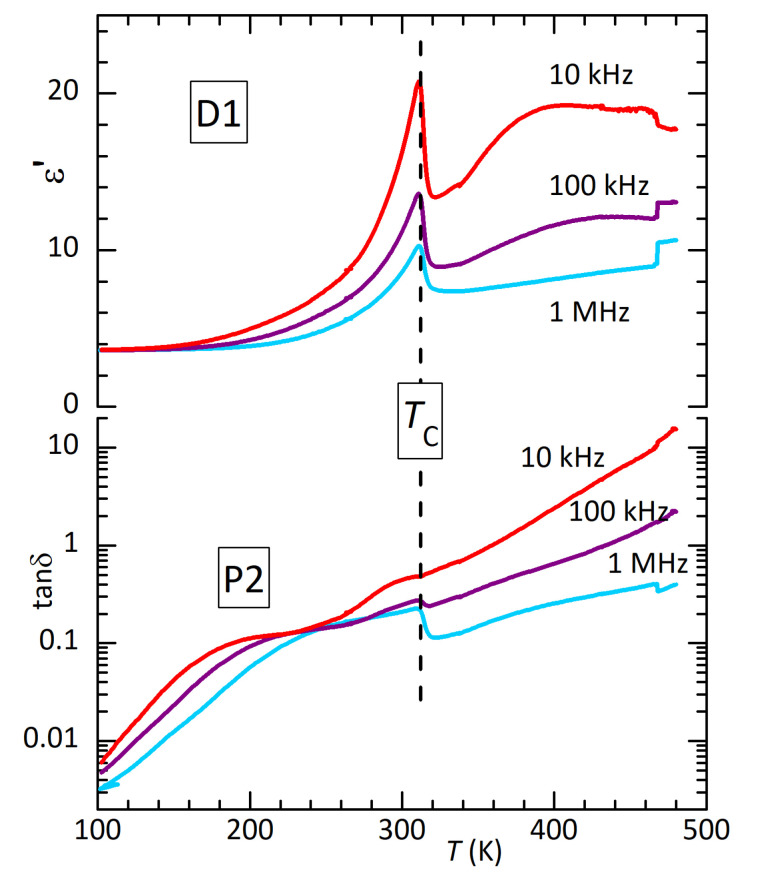
Dielectric permittivity of sample D1 during cooling, after 490 K has been reached.

**Figure 6 materials-16-07323-f006:**
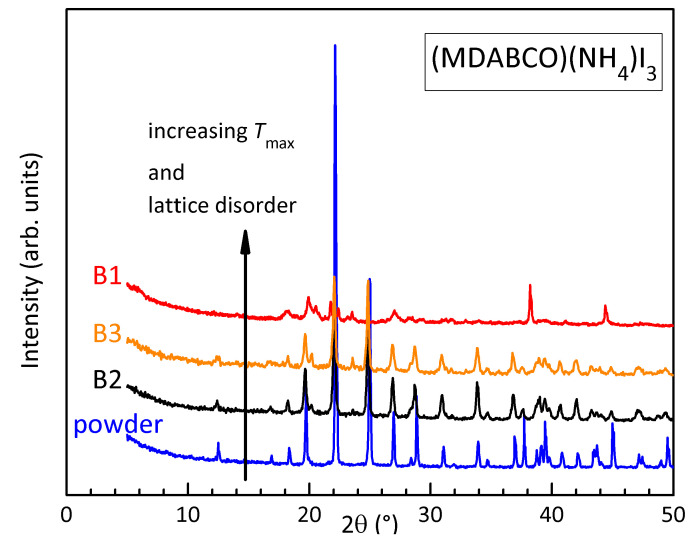
XRD diffractograms of the powder of (MDABCO)(NH${}_{4}$)I${}_{3}$ and of samples B1–B3 after the measurements in Figure 2.

**Figure 7 materials-16-07323-f007:**
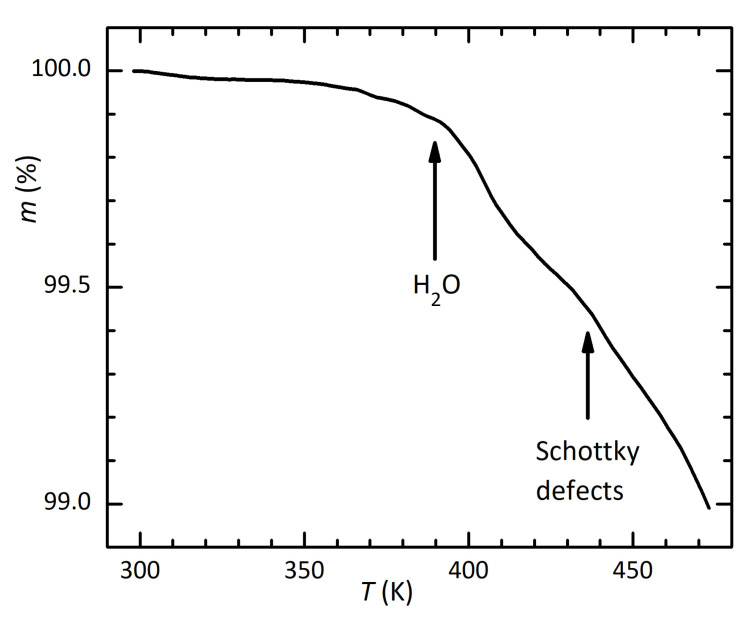
TGA of the powder of (MDABCO)(NH${}_{4}$)I${}_{3}$ at 5 K/min.

**Figure 8 materials-16-07323-f008:**
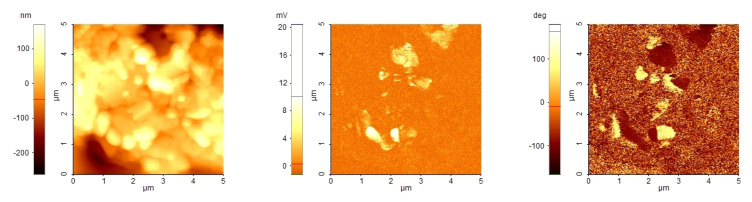
AFM, PFM amplitude and PFM phase of a (MDABCO)(NH${}_{4}$)I${}_{3}$ film deposited on ITO.

**Figure 9 materials-16-07323-f009:**
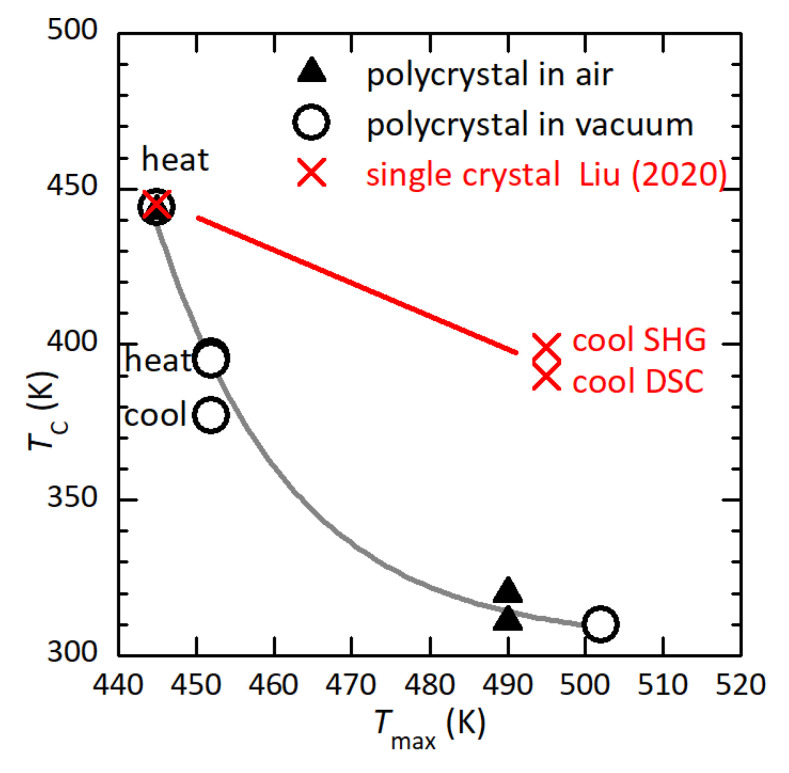
Transition temperature $T_{C}$ after having reached $T_{\max}$ in air or high vacuum. Shown in red are data from single crystals [2].

**Figure 10 materials-16-07323-f010:**
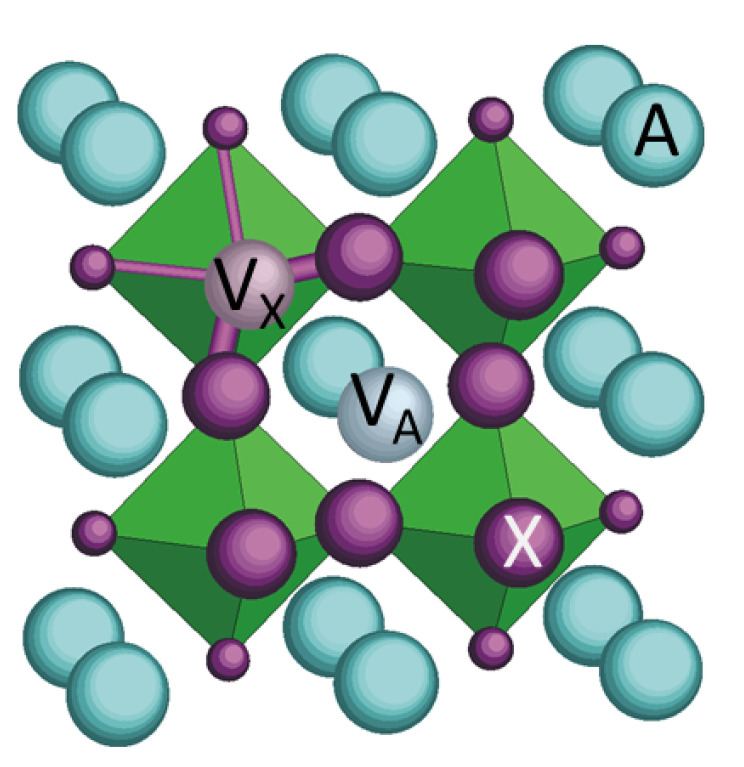
Section of the ABX${}_{3}$ perovskite lattice passing through a pair of A and X vacancies. The B atoms/molecules are at the centre of the octahedra. The X sites that are the nearest neighbour to V${}_{A}$ are drawn with a larger size than the others.

**Figure 11 materials-16-07323-f011:**
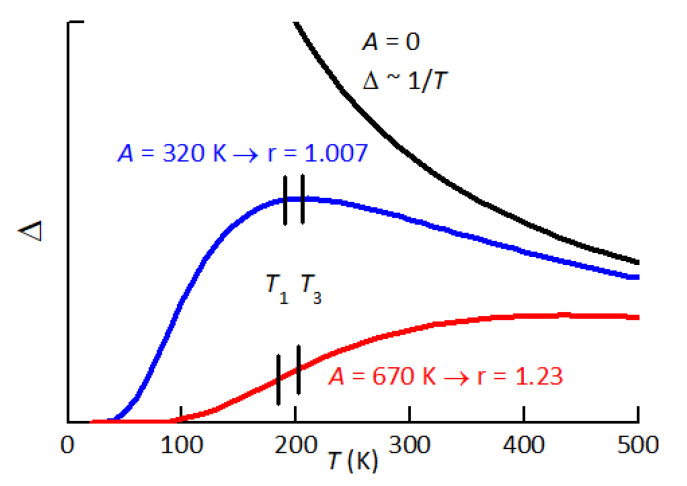
Temperature dependence of the relaxation strength between states differing in energy of *A*. The two curves with A=320 K and 670 K would explain the observed behaviour of peak P2 at low and high defect concentrations.

**Figure 12 materials-16-07323-f012:**
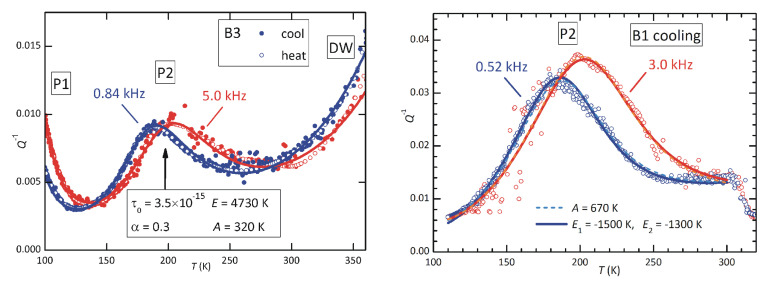
Fits of the anelastic spectra at low ((**left panel**) sample B3) and high ((**right panel**) sample B1) concentrations of defects. The fitting formulas and parameters are indicated in the text. The two fits of P1, dashed and continuous lines, are nearly coincident.

**Figure 13 materials-16-07323-f013:**
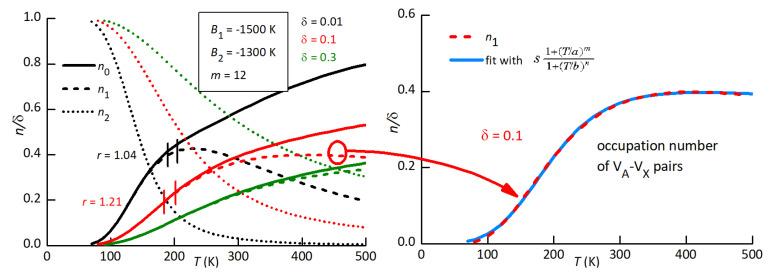
Populations $n_{0}$ of isolated V${}_{X}$, $n_{1}$ of V${}_{A}$–V${}_{X}$ pairs, $n_{2}$ of V${}_{X}$–V${}_{A}$–V${}_{X}$ complexes with the indicated parameter values. The dashed lines in the right panel are fits of the $n_{1}$ curves at δ=0.01 and 0.1 with the indicated rational expression.

**Table 1 materials-16-07323-t001:** Parameters used in the fits of Figure 12.

Sample	B3	B1	B1
*A* (K)	320	670	0
$B_{1}$ (K)	-	-	1500
$B_{2}$ (K)	-	-	1300
$\tau_{0}$ (s)	$3.5\times 10^{-15}$	$1.2\times 10^{-13}$	$3.4\times 10^{-12}$
*E* (K)	4730	4500	3600
α	0.3	0.1	0.15
β	0.3	0.6	0.6

## Data Availability

Data are contained within the article.

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
