# Peer review of "Phase Transition and Point Defects in the Ferroelectric Molecular Perovskite (MDABCO)(NH4)I3"

_materials, 2023, doi:10.3390/ma16237323_

Round 1

Reviewer 1 Report

Comments and Suggestions for Authors

This paper presents elastic and dielectric data obtained from a member of a relatively new class of halide perovskites in which both the A- and B-site cations are organic molecules, in place of metal cations. The piezoelectric or other physical properties of the phase investigated may or may not turn out to be valuable in practical applications but, at this stage, the focus is correctly on characterising their variations and stability. A high degree of variability between samples was attributed to non-homogeneous density within the pellets.

The bulk of the discussion relates to possible loss mechanisms, some of which may be intrinsic and some are associated with loss of crystallinity due to heating above ~400 K. A difficulty with such treatments is always that the interpretation relies on determinations of relaxation times and activation energies which can have a wide variety of origins. The most important unambiguous observations are, perhaps, that the polycrystalline pellets lose organic components when heated, and that cooling through the phase transition results in elastic stiffening rather than the softening observed in classic displacive phase transitions of oxide perovskites.

The quality of English is not ideal but the intended meaning is always clear enough. By way of a constructive criticism, it is not really sufficient to discuss the phase transition as being improper ferroelectric (line 7) without specifying the symmetry change. Classifying a phase transition as being proper, improper, ferroelastic, ferroelectric, etc., depends only on symmetry and is independent of whatever crystal structure of change in structure might be involved. The authors presumably know the space groups of the parent and product structures and these should be given. They can then specify the nature of the transition. The dynamic response of the structure to an applied stress, however, depends on the relaxation time of the order parameter which is a property of the individual crystal structure under consideration. Likewise, domain wall relaxation is discussed in relation to the acoustic loss (line 93) without specifying what type of domain walls these might be. Presumably the phase transition is ferroelastic but is it improper ferroelastic? This issue would be clear if the relevant space groups were specified. 

Comments on the Quality of English Language

The English is not perfect but the intended meaning is always clear.

Author Response

This paper presents elastic and dielectric data obtained from a member of a relatively new class of halide perovskites in which both the A- and B-site cations are organic molecules, in place of metal cations. The piezoelectric or other physical properties of the phase investigated may or may not turn out to be valuable in practical applications but, at this stage, the focus is correctly on characterising their variations and stability. A high degree of variability between samples was attributed to non-homogeneous density within the pellets.

The bulk of the discussion relates to possible loss mechanisms, some of which may be intrinsic and some are associated with loss of crystallinity due to heating above ~400 K. A difficulty with such treatments is always that the interpretation relies on determinations of relaxation times and activation energies which can have a wide variety of origins. The most important unambiguous observations are, perhaps, that the polycrystalline pellets lose organic components when heated, and that cooling through the phase transition results in elastic stiffening rather than the softening observed in classic displacive phase transitions of oxide perovskites.

The quality of English is not ideal but the intended meaning is always clear enough. By way of a constructive criticism, it is not really sufficient to discuss the phase transition as being improper ferroelectric (line 7) without specifying the symmetry change. Classifying a phase transition as being proper, improper, ferroelastic, ferroelectric, etc., depends only on symmetry and is independent of whatever crystal structure of change in structure might be involved. The authors presumably know the space groups of the parent and product structures and these should be given. They can then specify the nature of the transition. The dynamic response of the structure to an applied stress, however, depends on the relaxation time of the order parameter which is a property of the individual crystal structure under consideration. Likewise, domain wall relaxation is discussed in relation to the acoustic loss (line 93) without specifying what type of domain walls these might be. Presumably the phase transition is ferroelastic but is it improper ferroelastic? This issue would be clear if the relevant space groups were specified. 

Reply: we used XRD only for checking the quality of the synthesized powder and qualitatively assessing the deterioration of crystallinity after reaching high temperatures. Therefore, we rely on the literature for the space groups of the high and low temperature phases. Section 4.2 has been considerably expanded and mostly rewritten, describing in more detail the nature of the transition, as reported in literature for single crystals. The observation of stiffening below the transition is not compatible with a purely proper ferroelectric transition, regardless of the space groups of the two phases. In fact, it implies that the primary order parameter is not the electric polarization, which should be quadratically coupled with strain, and hence produce softening.

In view of the seeming inconsistency between our observations and the nature of the transition reported for crystals, we added considerations on the fact that our observations are not a consequence of a low quality of the samples.

Regarding the domain walls, we suppose that they are the usual walls between the eight possible orientations of the polar axis in rhombohedral ferroelectrics, but since we do not have indications whether the prevalent type of non-180o walls is 71o or 109o, we do not go beyond saying that the relaxation should be due to the motion of domain walls.

Other changes

We modified the reference to TC = 448 K of Ref. [11] into TC = 446 K. In fact, in that paper it is defined T0 = 448 K as "upper transition temperature" and the “Phase transition Temperature”, namely TC , as 446 K in the Supplemental Material, Table S10.

In this manner we make more compelling the close correspondence of our TC, defined as half the step of the Young's modulus, and the literature value of the single crystal.

Reviewer 2 Report

Comments and Suggestions for Authors

Author Response

Comments to the authors:

The authors have studied the anelastic, dielectric and structural properties of the metal-free molecular perovskite (ABX3) (MDABCO)(NH4)I3. They discuss the defect structure as a function of temperature and provide information on the process of thermal decomposition that occurs near TC. The authors identify the main mechanism of degradation as the formation of Schottky defects. A large part of the article is devoted to the discussion and mathematical interpretation of the defect structure.

The article is well written and understandable and the figures are well presented, but it leaves some minor comments.

Questions and comments that arise are the following:

- Please add “Curie temperature TC=448K” as a definition in the abstract.

Reply: We prefer avoid using the term Curie temperature, in view of the character of the phase transition. After our measurements it is not a proper ferroelectric transition.

- It would be better to change Chapter 4 “Materials and methods” before the results. All data should be discussed before the results as the reader can follow the discussion better. The order is so more logical.

Reply: “Materials and methods” is now before “Results”

- Figure 1: The curves are somewhat difficult to differentiate clearly.

Reply: the figure is the results of a long series of attempts to put the all the curves discussed in the text on a same plot. In fact, we find much more informative to compare the curves on a same plot rather than in separate plots. We enlarged of 25% the figure, so that the effectively confusing Q-1 curves near Tc can be better distinguished.

- Line 373: Is there a way to make the powder more homogeneous?

Reply: we tried many methods. The usual method of tapping the die on the table is not effective with such long shapes. We prepared a tool to be passed over the surface of the powder to make it flat. Then, we push very slowly the upper plug, in order to avoid that air turbulence moves the powder. Afterwards we tap repeatedly the dye. We suspect that the real problem is that the powder may be electrostatically attracted as the upper plug approaches and therefore tried to put glossy paper over the powder and flatten it before inserting the plug. Unfortunately, wrinkles form in the paper during pressing, that cause thin grooves on the sample surface and enhance its fragility.

- Figure 1, Lines 67-84: What influence on the curves would have to be mentioned in detail? Is it possible to give a statistical error based on repeated trials?

Reply: we only measured three samples, and therefore it would be improper to give a statistical error. The influence on the E(T) curves is only a multiplicative factor.

We prepared also the two figures below, which however are less clear than the original one. If the Reviewer thinks it necessary, we may substitute the original Fig. 1 with Fig. 1b below, or add Fig. 1c or only its upper panel.

Fig. 1b                                                                                                    Fig. 1c

In Fig. 1b an error bar for the initial value of E is added. Its extremes correspond to initial E/E0 of B1 and B2, adopting the same E0 = 7.4 GPa of B3. In Fig. 1c the absolute E is plotted instead of E/E0, but it is confusing, because the differences are not real and only due to wrong multiplication factors from the inhomogeneous densities of the samples. One also loses the repeatability of the initial normalized E curves and the correspondence between the magnitude of unrecoverable softening and the maximum temperature reached.

- Line 388: How homogeneous are the film thicknesses in the drop process? Is spin coating better?

Reply: the films thicknesses were quite inhomogeneous, with variations of the order of several microns, and probably spin coating would be better, but the reason why we also tested the films was that we wanted to verify the existence of a piezoelectric response. In fact, even if the XRD spectra were as expected, the dielectric response was vitiated by a large conductivity, presumably intergranular, and the elastic anomaly was not as expected from a proper ferroelectric, so that we doubted that there might be some problem with the synthesis of the material. The PFM technique allows the piezoelectric response to be probed at the domain level, so getting rid of the intergranular conductivity, and indeed demonstrated the existence of a piezoelectric response.

- Figure 5: What does "?" mean in P2?

Reply: We deleted the question mark. In the comment to Fig. 5 is written: “The thermally activated maximum around 200 K in tand seems compatible with P2, but the steep background makes any quantitative comparison or analysis difficult”. We are not completely sure that the maximum in e’ is the dielectric counterpart of anelastic P2, but it is very likely, since a pair of cation and anion vacancies has a strong electric dipole, and the temperature range of the dielectric maximum is compatible with that of the anelastic peak at lower frequency.

Other changes

We modified the reference to TC = 448 K of Ref. [11] into TC = 446 K. In fact, in that paper it is defined T0 = 448 K as "upper transition temperature" and the “Phase transition Temperature”, namely TC , as 446 K in the Supplemental Material, Table S10.

In this manner we make more compelling the close correspondence of our TC, defined as half the step of the Young's modulus, and the literature value of the single crystal.

Reviewer 3 Report

Comments and Suggestions for Authors

 The article provides a comprehensive overview of the phase transition and defect formation in organic perovskite. The authors have conducted a careful analysis of defect behaviors during the temperature-induced degradation of (MDABCO)(NH4)I3. In general, the paper is well-written and is suitable for publication with minor revisions.

1.The observed degradation of organic perovskite raises concerns regarding its potential application as a piezoelectric material. The pronounced degradation demonstrated in the study questions the practical utility of materials of this type. It is advisable for the authors to include additional discussions addressing this issue.

2. The authors have presented a discussion on defect behavior. I recommend that the authors also provide further insights into the comparable concentrations of Va and Vb defects.

Author Response

The article provides a comprehensive overview of the phase transition and defect formation in organic perovskite. The authors have conducted a careful analysis of defect behaviors during the temperature-induced degradation of (MDABCO)(NH4)I3. In general, the paper is well-written and is suitable for publication with minor revisions.

1.The observed degradation of organic perovskite raises concerns regarding its potential application as a piezoelectric material. The pronounced degradation demonstrated in the study questions the practical utility of materials of this type. It is advisable for the authors to include additional discussions addressing this issue.

Reply: this material deteriorates already at TC because its TC is particularly high, close to the onset of decomposition of any organic material. Therefore, it is not less suitable for applications than other organic ferroelectrics. We added a sentence at the end of Sect. 4.1.

  1. The authors have presented a discussion on defect behavior. I recommend that the authors also provide further insights into the comparable concentrations of Va and Vb defects.

Reply: Additional considerations on the relative concentrations of VA and VB would be purely speculative. In fact, we suggest but not prove that peak P1 is due to VB-VI pairs, and we were able to detect the tail of P1 in only one measurement. Therefore, we do not have reliable indications on its intensity and dependence on the total defect concentration.

Yes

Can be improved

Must be improved

Not applicable

Does the introduction provide sufficient background and include all relevant references?

( )

( )

(x)

( )

Are all the cited references relevant to the research?

( )

(x)

( )

( )

Is the research design appropriate?

(x)

( )

( )

( )

Are the methods adequately described?

( )

(x)

( )

( )

Are the results clearly presented?

( )

(x)

( )

( )

Are the conclusions supported by the results?

(x)

( )

( )

( )

The Introduction has been expanded, adding Refs. 2-10.

Other changes

We modified the reference to TC = 448 K of Ref. [11] into TC = 446 K. In fact, in that paper it is defined T0 = 448 K as "upper transition temperature" and the “Phase transition Temperature”, namely TC , as 446 K in the Supplemental Material, Table S10.

In this manner we make more compelling the close correspondence of our TC, defined as half the step of the Young's modulus, and the literature value of the single crystal.
